# Separation and Detection of Abamectin, Ivermectin, Albendazole and Three Metabolites in Eggs Using Reversed-Phase HPLC Coupled with a Photo Diode Array Detector

**DOI:** 10.3390/foods11233894

**Published:** 2022-12-02

**Authors:** Yawen Guo, Zhaoyuan He, Yali Zhu, Shuyu Liu, Pengfei Gao, Kaizhou Xie, Tao Zhang, Yuhao Dong

**Affiliations:** 1College of Animal Science and Technology, Yangzhou University, Yangzhou 225009, China; 2Joint International Research Laboratory of Agriculture & Agri-Product Safety, Yangzhou University, Yangzhou 225009, China; 3College of Animal Science and Technology, Guangxi University, Nanning 530000, China; 4College of Veterinary Medicine, Nanjing Agricultural University, Nanjing 210095, China

**Keywords:** abamectin, ivermectin, albendazole, HPLC–PDAD, LLE

## Abstract

An innovative and sensitive approach using high-performance liquid chromatography-photo diode array detection (HPLC-PDAD) was developed and optimized for the simultaneous determination of abamectin (ABM), ivermectin (IVM), albendazole (ABZ) and three metabolites in eggs. The samples were extracted with acetonitrile (MeCN)/water (90:10, *v*/*v*), and the extracts containing the targets were cleaned up and concentrated by a series of liquid–liquid extraction (LLE) steps. A reversed-phase C18 column and a mobile phase consisting of 0.1% trifluoroacetic acid (TFA) aqueous solution and methanol (MeOH) were utilized to perform optimal chromatographic separation. The developed method was validated on the basis of international guidelines. The limits of detection (LODs) and quantitation (LOQs) were 2.1–10.5 µg/kg and 7.8–28.4 µg/kg, respectively. Satisfactory linear relationships were observed for the targets in their corresponding concentration ranges. The mean recoveries ranged from 85.7% to 97.21% at 4 addition levels, with intraday and interday relative standard deviations (RSDs) in the ranges of 1.68–4.77% and 1.74–5.31%, respectively. The presented protocol was demonstrated to be applicable and reliable by being applied for the detection of target residues in locally sourced egg samples.

## 1. Introduction

ABM, produced by the fermentation of *Streptomyces avermectinius*, is a macrolide antibiotic that has a powerful repellent and killing effect on nematodes and arthropod species. Since its introduction, it has reduced the dose of antiparasitic drugs from the mg/kg level to the μg/kg level. ABM can bind tightly with soil and does not easily scour or infiltrate. ABM can be degraded into inactive compounds under light irradiation or under the action of soil microorganisms, and can ultimately act as a carbon source for plants and microbial decomposition [1]. Therefore, ABM has slight environmental residual toxicity and high safety; it has been extended to agricultural pest control and has a wide range of applications. ABM interacts stereoselectively and has a high affinity for the parasite glutamate-gated chloride channel; the subsequent chloride ion flux into neurons causes nerve conduction to be blocked, which presumably leads to paralysis and death of the parasite [2].

ABM and IVM are the most frequently administered members of the avermectin (AVM) family. AVMs are generally recommended to be administered once every 30 days in the form of feed supplements or by water addition in industrial henneries [3]. As early as 2011, the sales of IVM alone (used in animal health) exceeded US$ 1 billion per annum [3]. The main sites of biotransformation of AVMs in the body are the liver and fat, where the concentration is highest, and the elimination rate is slowest. In addition to liver and fat, residues may accumulate in kidneys, muscles, and eggs. As the most potent avermectin, IVM has similar antiparasitic activity as ABM, yet it has stronger penetration into mammalian tissues and can penetrate organs and muscles that are difficult to reach by general antiparasitic drugs. The validity period is prolonged, and the safety is improved. The antiparasitic mechanism of IVM involves binding to ligand-gated ion channels and receptors, including glutamate-gated Cl channels, G-protein-gated inwardly rectifying K channels, gamma aminobutyric acid type-A receptors, cysteine-loop receptors, glycine receptors, purinergic P2X4 receptors, and fernesoid X receptors, resulting in parasite and parasite vector paralysis and death [3]. IVM has a prominent effect on gastrointestinal parasitic nematodes, lung flukes, lice, and scabies mites, yet it is insufficient for tapeworms, trematodes and protozoa, not against avian *Heterakis*.

ABZ is a methylcarbamate benzimidazole antiparasitic compound that has a broad spectrum and is effective against lungworms, tapeworms, gastrointestinal nematodes, and flukes. ABZ acts by binding to β-tubulin and inhibiting its polymerization to microtubules, resulting in impaired energy metabolism in the parasite [4]. In actual production, laying hens are frequently supplemented with a combination of ABZ and IVM at a ratio of 50:1 before the start of production to repel various parasites in vivo and in vitro [4]. Mature layers are treated against helminths with a combination of ABZ and IVM administered twice, with an interval of 20–30 days [4], preferably in the summer and autumn when parasites are abundantly reproducing.

Chemotherapy is the most direct and widely used method to control parasitic infections. Currently, antiparasitic combinations, including ABM, IVM, ABZ, and other components, are common in the veterinary pharmaceutical market [5]. Although ABM and IVM are administered in smaller doses, at the μg/kg and ng/kg levels, both drugs are highly liposoluble, have a long residual time in animals and are neurotoxic, developmental, and clastogenic toxicities [6]. Zebrafish embryo exposure experiments confirmed that the developmental toxicity and teratogenic effects were displayed by ABZ itself, rather than its first metabolite albendazole sulfoxide (ABZSO_2_) and the subsequent main metabolites albendazole sulfone (ABZSO) and albendazole-2-aminosulfone (ABZSO_2_NH_2_) [7]. It has been proven that egg hatchability was decreased when the dose of ABZ in medicated feed was 40–80 mg/kg [8]. Residues in foods of animal origin at low concentrations are unlikely to lead directly to the toxic effects described above. However, there are potential hazards to humans due to food chain accumulation and possible overuse in food animal production systems. The accompanying development of resistance is difficult to monitor and poses a threat to public health [9]. Herein, the proposal and optimization of simultaneous detection methods for ABM, IVM, ABZ, and three metabolites are in demand for safeguarding food safety.

Maximum residue limits (MRLs) for ABM, IVM and ABZ in poultry and eggs are not stipulated by the EU [10] and China [11]. The MRLs of ABM are 100 μg/kg in EU for ovine muscle and in China for bovine fat and liver, the MRLs of IVM are 30 μg/kg in EU for kidney of food animals and in China for muscles of cattle, sheep, and pigs, and the MRLs of ABZ are 100 μg/kg in EU for muscle of ruminants and in China for muscle of food animals. In this study, 100 μg/kg, 30 μg/kg and 100 μg/kg were adopted as the MRLs of ABM, IVM and ABZ (the sum of ABZ and three metabolites) in eggs, respectively. On this basis, the objective of this study was to establish an unreported and innovative HPLC-PDAD method for the simultaneous detection of ABM, IVM, ABZ and three metabolites.

## 2. Materials and Methods

### 2.1. Chemicals and Reagents

The ABM reference standard (CAS No. 71751−41−2, purity ≥ 98.0%, *w*/*w* of ABM B_1a_ > 90% and *w*/*w* of ABM B_1b_ < 5%) and IVM reference standard (CAS No. 70288-86-7, purity ≥ 98.0%, *w*/*w* of IVM B_1a_ > 85% and *w*/*w* of IVM B_1b_ > 8%) were obtained from Yuanye Bio−Technology Co., Ltd. (Shanghai, China). ABZ reference standard (CAS No. 54965-21-8, purity ≥ 98.0%), ABZSO_2_ reference standard (CAS No. 75184-71-3, purity ≥ 98.0%), ABZSO reference standard (CAS No. 54029-12-8, purity ≥ 98.0%) and ABZSO_2_NH_2_ reference standard (CAS No. 80983-34-2, purity ≥ 99.8%) were acquired from Sigma–Aldrich LLC (St. Louis, MO, USA). MeOH was of HPLC grade and was provided by Tedia Company Inc. (Fairfield, OH, USA). TFA was of HPLC grade and was supplied by Aladdin Reagent Co., Ltd. (Shanghai, China). MeCN was of HPLC grade and was purchased from Sinopharm Chemical Reagent Co., Ltd. (Shanghai, China). The water (18.25 MΩ*cm, 25 °C) used in the experiment was purified and generated by an automatic water purification system (Milli–Q HR 7000, Merck Drugs & Biotechnology Co., Inc., Fairfield, OH, USA) equipped with a dual wavelength ultraviolet lamp for sterilization. The solution entered into the HPLC system was first degassed by an ultrasonic apparatus (P300H, Elma Electronic GmbH, Stuttgart, Germany).

### 2.2. Preparation of the Standard Stock and Working Solutions

The standard stock solutions of ABM, IVM, ABZ, ABZSO_2_, ABZSO, and ABZSO_2_NH_2_ were prepared individually at a concentration of 1 mg/mL by dissolving each target in MeOH and then stored stably for two months in actinic glassware at −18 °C. The standard working solutions were prepared daily by gradually diluting the standard stock solutions with MeOH.

### 2.3. Sample Acquisition and Preparation

Eggs without any detectable targets were collected from 30-week-old Haiyang yellow chickens (Jinghai Poultry Industry Group Co., Ltd., Nantong, China) after 2 weeks of feeding with nonmedicated feed. After the eggs with broken eggshells were discarded, the whole egg, egg yolk and egg white samples were separated and fully homogenized. Whole egg, egg yolk and egg white were regarded as different sample substrates because the yolk had a longer development period than the egg white during egg formation, and different drugs have been shown to have different metabolic transformations and substance binding abilities in the egg yolk and white [9]. In addition, the current consumer market has a more refined demand for egg yolk and egg white, such as cakes and fitness meals, which have corresponding demands for the yolk and white, respectively.

The homogenized egg sample (2.00 ± 0.01 g) was precisely weighed in a 50 mL centrifuge tube, and then 5 mL of MeCN/water (90:10, *v*/*v*) was added. The mixture was stirred on a vortex mixer (Vortex-Genie 2, Scientific Industries Inc., Bohemia, NY, USA) for 3 min, followed by ultrasonic homogenization using an ultrasonic cleaner (P300H, Elma Electronic Ltd., Munich, Germany) for 10 min. After centrifugation at 8000 rpm for 10 min, the supernatant was transferred to another centrifuge tube, and the extraction step was repeated twice to combine the two supernatants. Subsequently, LLE and Quick, Easy, Cheap, Effective, Rugged, and Safe (QuEChERS) extraction were used in the extraction process.

#### 2.3.1. LLE

The pooled supernatants were evaporated to near dryness (1–2 mL) in a nitrogen blow concentrator (N–EVAP–24, Organomation Associates Inc., Berlin, MA, USA) in a fume hood, and the needle height was adjusted as the liquid level dropped. The temperature of the aluminum bead bath was set to 40 °C, and the steady nitrogen stream was supplied by a nitrogen generator (Genius 1024, Peak Scientific Instruments Ltd., Inchinnan, UK). The residue obtained in the prior step was dissolved by adding 2 mL MeCN/water (90:10, *v*/*v*) and stirring for 3 min. Then, 5 mL of n-hexane saturated with ethyl acetate was added to the egg yolk and total egg samples and stirred for 2 min to defatten the samples; the volume added to the egg white samples was 2 mL. After the fat layer was removed, the solution was concentrated to near dryness in a nitrogen blow concentrator. The residue was resuspended in 2 mL of the initial mobile phase (0.1% TFA aqueous/MeOH, 57:43, *v*/*v*), stirred for 2 min, and injected through a 0.22 μm organic-phase sterile needle filter into the LC autosampler vial.

#### 2.3.2. QuEChERS Extraction

QuEChERS scavenger (4 g of anhydrous magnesium sulfate, 1 g of sodium chloride, and 1.5 g of sodium citrate buffer) was added to the pooled supernatants, stirred for 2 min, and centrifuged at 10,000 rpm for 10 min. The residue concentrated by the nitrogen blow concentrator was reconstituted in 2 mL of initial mobile phase, stirred for 2 min, and poured through a 0.22 μm organic-phase needle filter.

### 2.4. Instruments and Conditions

LC separation was performed on an Alliance e2695 HPLC system (Waters Corp., Milford, MA, USA), detection was achieved on a 2998 PDA detector (Waters Corp., Milford, MA, USA), and retention of the targets was accomplished by a Waters XBridge BEH C18 column (4.6 mm × 150 mm, 5 µm). Instrument linkage and condition control were conducted on Empower 3 software (Waters Corp., Milford, MA, USA). The column oven temperature was held at 30 °C, and the injection volume was 10 µL. The mobile phase consisted of 0.1% TFA aqueous solution (component A) and MeOH (component B) at a flow rate of 1.0 mL/min. The gradient elution process was initiated with a 57:43 ratio of component A and component B, lasting for 4 min, and the detection wavelength during this period was set to 288 nm. The gradient elution procedure was as follows: 4–6.6 min, 100% B, 302 nm; 6.6–8.5 min, 100% B, 244 nm; 8.5–10 min, 100% B, 288 nm; and 10–11 min, 43% B, 288 nm.

### 2.5. Method Parameters

The validation parameters of the analytical method followed the corresponding guidelines of the European Commission [12] and the U.S. Food and Drug Administration [13].

The lowest target concentration that can be detected is the LOD, and the lowest concentration that can be quantified is the LOQ. LOD and LOQ were calculated using the signal-to-noise ratio (S/N) method to assess the sensitivity, and when S/N ≥ 3 and 10, the corresponding actual spiked concentrations were LOD and LOQ, respectively. The height of the target peak is the signal, and the height of the selected relatively smooth baseline is the noise.

The initial mobile phase was used to prepare mixed standard working solutions by diluting the standard stock solutions of six targets, with concentration gradients of LOQ, 50.0, 80.0, 100.0, 200.0, 400.0, and 600.0 μg/L for ABM; LOQ, 30.0, 60.0, 100.0, 120.0, 200.0, and 400.0 μg/L for IVM; and LOQ, 30.0, 50.0, 100.0, 200.0, 400.0, and 600.0 μg/L for ABZSO_2_NH_2_. The concentration gradients of ABZ, ABZSO_2_ and ABZSO started from the respective LOQs, followed by 50.0, 80.0, 100.0, 200.0, 400.0, and 600.0 μg/L. The established HPLC-PDAD method was used to detect the abovementioned mixed standard working solution, and the standard curve was plotted with the peak area on the *y*–axis and the actual concentration of each target on the *x*–axis.

The added concentrations of the six targets in the 2.00 g homogenized sample were LOQ, 0.5 MRL, 1.0 MRL, and 2.0 MRL, each concentration contained six parallels, the detected peak area was substituted into the standard curve to calculate the detected concentration, and the ratio of the detected concentration to the actual added concentration was the recovery. The precision of the method was measured by the intraassay RSD and interassay RSD. Samples with four added levels (LOQ, 0.5 MRL, 1.0 MRL, and 2.0 MRL) were detected with the same standard curve at three different times on the same day. Each concentration included six replicates, and the intraassay RSD was calculated. Samples with four added concentrations, each containing six replicates, were detected on three different days of the week using different standard curves to obtain interassay RSDs.

Stability assessment of targets was carried out in MeOH and various sample substrates (whole egg, egg yolk and egg white) under different storage conditions. Standard stock solutions can remain steady for more than 2 months after repeated freeze–thaw cycles. Standard working solutions can remain stable for more than 2 weeks at 4 °C, and standard working solutions of different concentrations can remain stable for more than 2 days at room temperature (25 °C) when added to the sample substrates.

## 3. Results and Discussion

### 3.1. Optimization of Sample Preparation

The complex components in the sample can clog or contaminate the instrument and interfere with the chromatographic behavior of the targets. Sample preparation is the process of removing as many interfering components as possible, purifying impurities coextracted with the targets, and extracting and concentrating the targets from the sample matrix. In this work, two sample preparation strategies (LLE and QuEChERS) were evaluated based on recovery results. Magnesium sulfate was used in the QuEChERS strategy instead of sodium sulfate because the former has a stronger drying capacity to remove water from MeCN. The presence of water increases the polarity of MeCN, and incomplete extraction of low-polarity compounds, such as fats and proteins, interferes with the targets [14]. In general, the LLE procedure is cumbersome and time-consuming, and excessive transfer steps increase the risk of sample contamination and loss of targets [9]. Intriguingly, the data in Table 1 show a similar range of recoveries obtained for the six targets extracted from total egg and egg white using LLE and QuEChERS. However, for egg yolk, the range of recoveries obtained by LLE (88.61–93.85%) was higher than that obtained by QuEChERS (79.04–84.49%). There were significant differences in the recoveries of the four targets (ABM, IVM, ABZ, and ABZSO) in egg yolk between the two sample preparation methods (*p* < 0.05). Considering the expensive commercial products of QuEChERS, LLE was ultimately chosen as the sample preparation method in the proposed study.

In the LC-fluorescence detection (FLD) methods of ABZ and its three metabolites, the extractant used was ethyl acetate [4,15,16]. Preexperiments showed that the extraction capacity of ethyl acetate for ABM and IVM was insufficient. In the report by Danaher et al., acetone/water (1:1, *v*/*v*) and isooctane were added successively to extract four AVMs from bovine liver [17]. We did not consider this option because acetone is a laboratory-controlled reagent and isooctane is not commonly applied in egg samples. Comparatively, MeCN and MeOH were the common initial extractants [9,18,19,20,21], and the extracted solution was found to be turbid and oily when MeOH was used as the extractant. Both the study by Schenck and Lagman [19] and the research by Wang et al. [22] utilized MeCN alone to extract AVMs and obtained the desired recovery ranges. Zhang et al. used MeCN alone in the extraction of ten benzimidazoles (BMZs), including ABZ and three metabolites [21]. After comparison, it was found that the extraction efficiency of MeCN/water was higher than that of MeCN alone for the targets. In addition, we conducted parallel experiments on the extraction efficiency of MeCN/water with different volume ratios (80:20, 85:15, 90:10, and 95:5) and found that the change in volume ratio had a more obvious influence on the recoveries of ABM and IVM and that 90:10 was the optimal ratio. Although the addition of water to the extractant increased the recovery, the time of nitrogen evaporation (>3 h) was prolonged. Nevertheless, the nitrogen evaporation process has a relatively short duration and a more efficient concentration for the targets compared to centrifugal concentration.

### 3.2. Optimization of HPLC-PDAD

PDA detection differs from mass spectrometry in that the targets need to be completely separated by an LC system without interfering with each other. Herein, we focus on the LC methods as references in the selection and optimization of chromatographic column, mobile phase, elution procedure, detection wavelength, and various other HPLC–PDAD conditions. The octadecylsilyl column, also known as the C18 column, is a commonly used reversed-phase chromatographic column with stable and durable packing particles, acid and base resistance, and reliable chromatographic performance [22]. This phenomenon contributes to the retention and separation of targets with different polarities in this study, including nonpolar IVM and weakly polar ABM and ABZ.

According to the publications, 0.025 mol/L ammonium acetate [21], 0.1% formic acid [23,24,25], 0.1% acetic acid [26,27,28], 0.1% triethylamine [29], and 0.1% TFA [22] were utilized as component A; MeOH and MeCN were used as component B; and components A and B composed of different mobile phases were tested. In comparison, the use of the first four reagents did not result in ideal peaks for ABM and IVM, while 0.1% TFA resulted in sharp and symmetrical peaks for all targets, as presented in Figure 1, Figure 2 and Figure 3. The pH of the 0.1% TFA aqueous solution ensures constant ionization of targets with different p*K*a values [22]. In particular, the maximum ultraviolet (UV) absorption peak of TFA was below 200 nm, which was different from the detection wavelengths of the six targets, with little interference. The overall retention time of the target was relatively early, and the overall signal of the target was stronger when MeCN was used, yet the peaks of ABZ were prone to splitting, which may be related to the fact that ABZ itself has relatively few chromophores. When MeOH was used, the target peaks were more stable and sufficiently separated from each other.

Isocratic elution of components A and B at different ratios was attempted, and eventually, the peak separation of the targets was observed to be good, but the shapes were poor, and the analysis times exceeded 20 min, hampering the efficiency of the experimental analysis. Afterward, we optimized the gradient variation for this proposed study based on the elution procedure described in the work of Permana et al. [22]. When the ratio of component A was set between 50% and 70%, ABZ and three metabolites could be eluted. When the ratio of component A was reduced to 0–20%, ABM and IVM could be eluted, and the best results were obtained when the ratio was below 5%. After constant attempts and optimization, the gradient program described in Section 2.4 was finally determined. The target peaks were separated from each other, and the peak shape was symmetrical. Different flow rates were also compared, and 1.0 mL/min was selected after taking into account the running time of the chromatography and the stability of the system pressure.

The presence of conjugated saturated and unsaturated bonds and heteroatoms such as nitrogen, sulfur, and oxygen in the structures of the targets renders them UV active compounds [30]. Nevertheless, the spectral properties are not absolute and are constrained by the chemical circumstances [30]; thus, the absorption spectrum of the targets was captured under optimized chromatographic conditions. The results indicated that both ABM and IVM had strong absorption at 244.2 nm; ABZ had strong absorption at 204.1 nm, 230.0 nm and 302.3 nm; ABZSO_2_, ABZSO and ABZSO_2_NH_2_ had strong absorption at 220.0 nm and 290.0 nm, and the response values were higher at 220.0 nm than at 290.0 nm, but the ABZ peak was interfered by the solvent peak at 220.0 nm. Eventually, 244.0 nm was selected as the detection wavelength for ABM and IVM, 302.0 nm for ABZ, and 288.0 nm for ABZSO_2_, ABZSO and ABZSO_2_NH_2_ based on the response of the targets and the interference of the solvent peaks. The corresponding detection wavelength of the target is different from the related reports. The detection wavelengths of both ABM and ABZ were assigned at 210 nm by Ali et al. [29]. The detection wavelengths of both IVM and ABZ were configured at 245 nm by Waldia et al. [31]. The detection wavelengths of both IVM and ABZ were fixed to 292 nm by Pawar et al. [30]. Permana et al. set the detection wavelength of ABZ and two metabolites (ABZSO_2_ and ABZSO) at 290 nm and that of IVM at 245 nm [22].

### 3.3. Analytical Method Validation

In this study, LODs of 3.2–9.8 µg/kg and LOQs of 7.9–26.6 µg/kg in whole egg were achieved, as illustrated in Table 2. The acquired LODs range in egg yolk and egg white are 2.6–10.5 µg/kg and 2.1–8.6 µg/kg, respectively, and the achieved LOQs range in egg yolk and egg white are 8.1–28.4 µg/kg and 7.8–25.0 µg/kg, respectively.

The matrix-matched calibration curve was fitted from the target’s respective seven concentration points and responses. Linearity was satisfactory, with a coefficient of determination (R^2^) ≥ 0.9993, as enumerated in Table 3.

The established sample preparation method resulted in good recoveries, ranging from 87.10% to 94.62% in total egg (RSD < 3.82%), 85.70% to 95.97% (RSD < 4.05%) in egg yolk, and 88.38% to 97.21% (RSD < 3.91%) in egg white when analysis was performed at four added estimations. The intraday and interday precisions (<5.31%) demonstrated in Table 4 are acceptable and inside the acceptance criteria of the guidelines [12,13]. Consequently, the low variability across the four estimations proves the accuracy, consistency, and reproducibility of the new method.

### 3.4. Method Application

A total of seven brands of eggs were purchased from three local sources (large supermarkets, specialty stores and grocery stores), and five eggs from each brand were considered as individual samples, fully homogenized and later detected according to the proposed method. Notably, none of the samples contained quantifiable levels of the target residues. Application in real samples proves the availability and adaptability of the method.

### 3.5. Comparison with Reported Methods

Permana et al. presented an HPLC–UV approach to detect IVM, ABZ, ABZSO_2_, ABZSO, and doxycycline in rat plasma and organs [22]. IVM and ABZ in bovine and poultry-derived samples were detected by the micellar LC method by Pawar et al. [30]. Waldia et al. developed an HPLC–UV method for the detection of IVM and ABZ in tablet dosage form [31]. Ali et al. applied HPLC–PDAD for the detection of ABM, ABZ, levamisole hydrochloride and closantel in oral suspensions [29]. Among the studies applying LC methods for simultaneous detection of AVMs (including ABM and IVM) and BMZs (including ABZ and three metabolites), no residues in foods of animal origin were involved.

The LC methods for the detection of target residues related to this study in foods of animal origin are summarized in Table 5. LC methods for simultaneous detection of ABM and IVM rely primarily on fluorescence detection (FLD) [14,17,18,19,20]. In addition, the FLD method requires that the target be derivatized prior to chromatographic separation. In contrast, the application of LC–FLD for the detection of ABZ (or metabolites together) residues does not require a derivatization process [4,15,16,32]. Therefore, in this scenario, it is problematic to adopt FLD for the simultaneous detection of chromatographically separated AVMs and BMZs. The remaining studies used UVD or PDAD to detect ABZ (or together with metabolites) residues [21,23,25,26,27,28]. Compared to the investigations listed in Table 5, the targets involved in this study belong to two drug classes (AVMs and BMZs), the established HPLC–PDAD method is time saving (11 min), the sample preparation method is simple, the performance parameters have been validated and the applicability has been demonstrated. Furthermore, not only does the introduction of PDAD present a new method and a supplement to the existing detection scheme, but the diode array detector itself is less sensitive to flow rate and temperature fluctuations than a UV detector and is more suitable for gradient elution and chromatographic retention. The detection elements of the diode array detector form multiple channels work in parallel, which can detect the optical signals of all wavelengths separated by the grating and can obtain chromatograms of any wavelength, providing richer information for qualitative and quantitative analysis. In contrast, when a UV detector is used for quantitative analysis of target compounds, detection is performed at the wavelength of maximum absorption, resulting in weaker absorption or even no absorption of other components in that channel, and the results may be seriously biased.

The optimizations and improvements we have made in the sample preparation method are mainly based on the comparison between LLE and QuEChERS and the selection of extractants as described in Section 3.1. A comparison between the sample preparation method we have proposed and those in reported studies is displayed in Table 5. Various sample preparation methods have been established, such as QuEChERS extraction–purification [17,25], dispersed liquid-phase microextraction [24,26,28,32], solid-phase microextraction [23], LLE–purification [4,14,18,20] and LLE alone [15,16,19,21,27]. In comparison, the sample preparation method without a clean-up step proposed in this study is simple to operate, and the recovery range obtained is ideal.

## 4. Conclusions

In the proposed method, an LLE-HPLC-PDAD method for the simultaneous detection of two AVMs and four BMZs in eggs was developed and optimized with high efficiency, low cost, and low time consumption. Moreover, the LOD, LOQ, linearity, recovery and precision were validated to prove that the method is specific, reliable, and stable. The application in the routine analysis of real samples demonstrates the practicality of the method. In conclusion, the novel validated method achieves the simultaneous detection of ABM, IVM, ABZ and three metabolites in eggs using an HPLC-PDAD approach for the first time.

## Figures and Tables

**Figure 1 foods-11-03894-f001:**
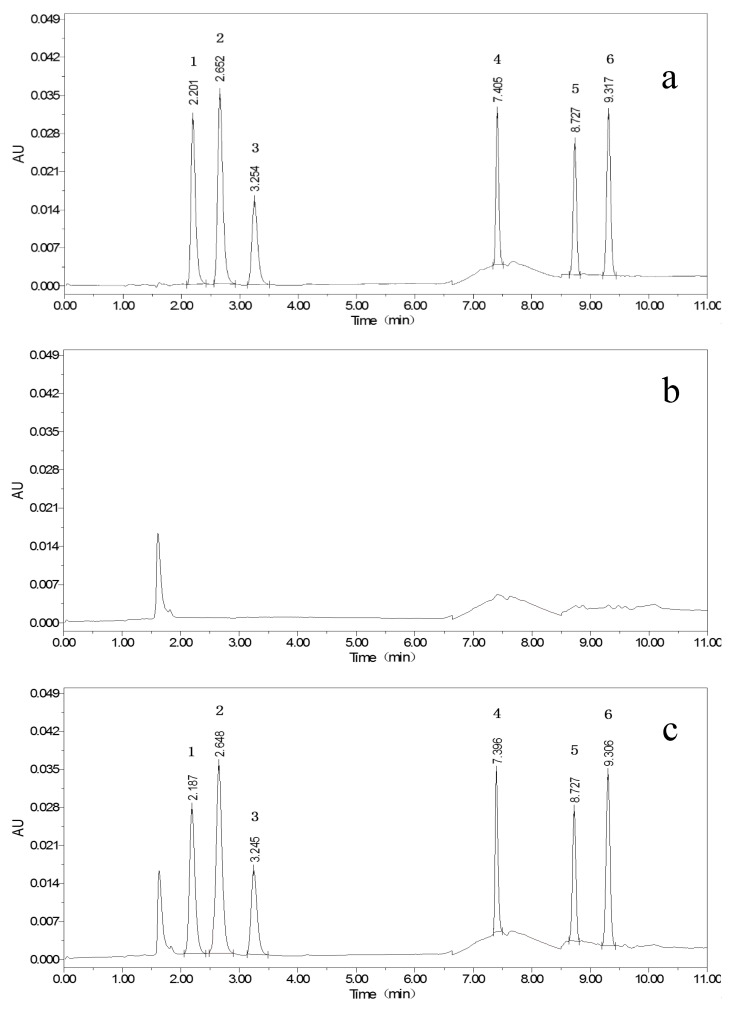
Chromatograms of 100 μg/kg mixed standards (**a**); blank whole egg (**b**); and blank whole egg with 100 μg/kg mixed standards (**c**) (ABZSO_2_NH_2_ (1), ABZSO (2), ABZSO_2_ (3), ABZ (4), AVM (5), IVM (6)).

**Figure 2 foods-11-03894-f002:**
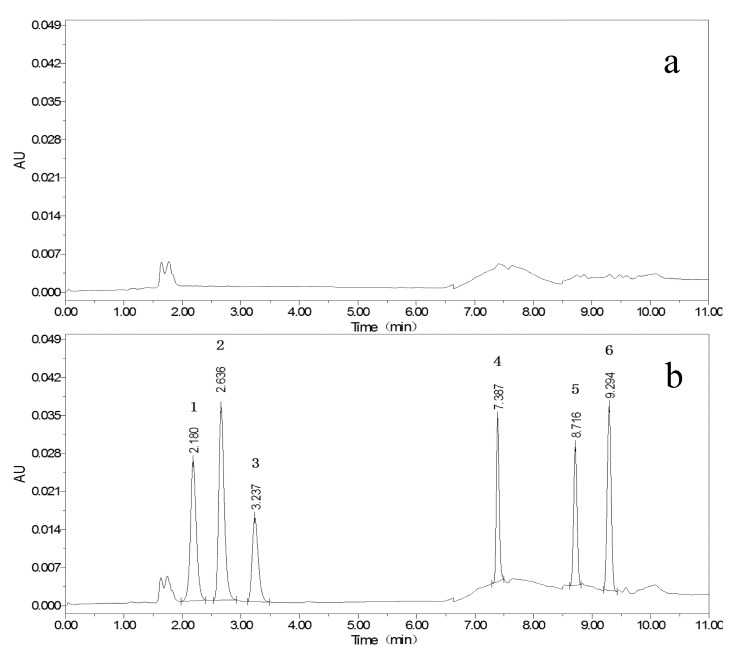
Chromatograms of blank egg yolk (**a**); and blank egg yolk with 100 μg/kg mixed standards (**b**) (ABZSO_2_NH_2_ (1), ABZSO (2), ABZSO_2_ (3), ABZ (4), AVM (5), IVM (6)).

**Figure 3 foods-11-03894-f003:**
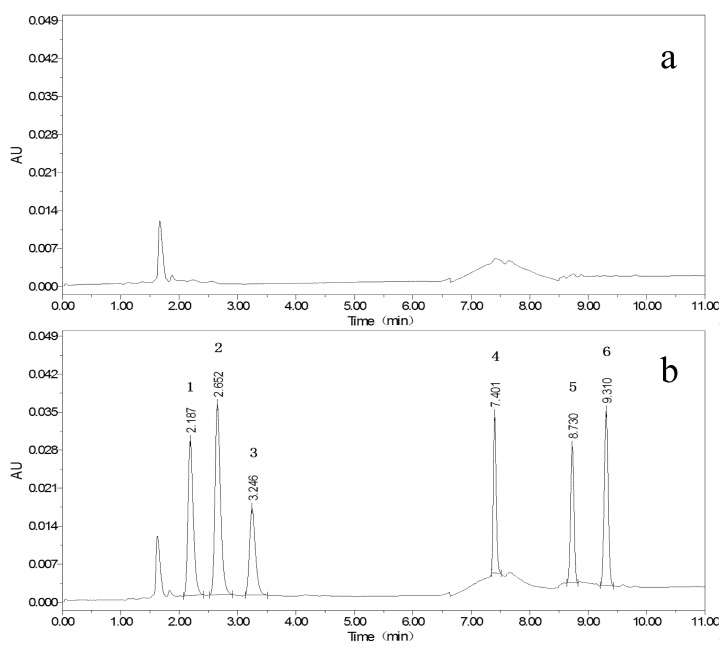
Chromatograms of blank egg white (**a**); and blank egg white with 100 μg/kg mixed standards (**b**) (ABZSO_2_NH_2_ (1), ABZSO (2), ABZSO_2_ (3), ABZ (4), AVM (5), IVM (6)).

**Table 1 foods-11-03894-t001:** Comparison of recoveries obtained from LLE and QuEChERS (%).

Analyte	Whole Egg		Egg Yolk		Egg White	
	LLE	QuEChERS	LLE	QuEChERS	LLE	QuEChERS
ABM	87.63 (3.37)	86.93 (2.23)	88.61 ^a^ (3.37)	79.04 ^b^ (2.17)	89.56 (2.85)	90.05 (1.99)
IVM	93.15 (2.28)	90.98 (2.35)	90.10 ^A^ (2.98)	81.91 ^B^ (2.50)	93.41 (2.23)	88.87 (1.77)
ABZ	92.82 (2.58)	89.15 (1.86)	90.78 ^a^ (2.37)	81.96 ^b^ (2.64)	93.03 (2.94)	92.64 (2.18)
ABZSO_2_	91.77 (2.73)	90.75 (2.09)	89.39 (2.45)	84.49 (2.05)	94.34 (3.08)	96.26 (1.98)
ABZSO	93.35 (2.51)	90.31 (2.33)	93.85 ^A^ (2.49)	82.83 ^B^ (2.61)	94.84 (2.99)	95.05 (2.21)
ABZSO_2_NH_2_	92.55 (2.59)	89.00 (1.96)	92.87 (2.98)	89.19 (2.54)	94.89 (2.97)	92.44 (2.51)

Note: *n* = 6, values in parentheses are SDs. ^a,b^ Means with different superscripts differ significantly (*p* < 0.05). ^A,B^ Means with different superscripts differ extremely significantly (*p* < 0.01).

**Table 2 foods-11-03894-t002:** LODs and LOQs of six target compounds in eggs (µg/kg).

Analyte	Whole Egg		Egg Yolk		Egg White	
	LOD	LOQ	LOD	LOQ	LOD	LOQ
ABM	8.5	26.0	8.6	26.4	7.9	25.0
IVM	3.5	10.6	3.0	11.4	2.8	9.5
ABZ	9.8	26.6	10.5	28.4	8.6	25.0
ABZSO_2_	6.6	21.3	7.8	22.1	6.0	20.0
ABZSO	3.2	7.9	2.6	8.1	2.1	7.8
ABZSO_2_NH_2_	3.5	21.4	2.8	10.6	3.6	11.5

Note: *n* = 3.

**Table 3 foods-11-03894-t003:** Linear regression equations, R^2^ values and linearity ranges of six targets.

Analyte	Slope	y-Intercept	R^2^	Linearity Range (µg/L)
ABM	1539	435.25	0.999 8	25.0–600.0
IVM	2259.2	−4382.5	0.999 6	9.5–400.0
ABZ	1477	−3598.5	0.999 6	25.0–600.0
ABZSO_2_	1606.1	3884.5	0.999 3	20.0–600.0
ABZSO	3359.1	−2829.4	0.999 4	7.8–600.0
ABZSO_2_NH_2_	2462.9	−5089.4	0.999 8	10.6–600.0

Note: *n* = 6.

**Table 4 foods-11-03894-t004:** Mean recoveries and precision of six targets.

Analyte	Whole Egg				Egg Yolk				Egg White			
	Added Level (µg/kg)	Recovery (%) ^α^	Intraday RSD (%) ^β^	Interday RSD (%) ^β^	Added Level (µg/kg)	Recovery (%) ^α^	Intraday RSD (%) ^β^	Interday RSD (%) ^β^	Added Level (µg/kg)	Recovery (%) ^α^	Intraday RSD (%) ^β^	Interday RSD (%) ^β^
ABM	26.0	87.10 (3.20)	3.06	4.14	26.4	85.70 (4.05)	3.05	4.82	25.0	88.38 (2.48)	2.51	2.85
	50	91.07 (3.27)	2.91	3.32	50	90.35 (3.34)	3.39	4.30	50	91.90 (3.15)	2.14	3.68
	100 ^γ^	91.06 (3.74)	3.05	3.76	100 ^γ^	88.94 (2.25)	2.33	3.71	100 ^γ^	92.21 (2.71)	2.54	3.84
	200	93.29 (3.25)	4.77	4.27	200	89.43 (3.85)	4.08	4.27	200	89.75 (3.05)	2.94	4.27
IVM	10.6	89.51 (3.07)	2.89	2.79	11.4	88.34 (2.69)	2.65	2.99	9.5	90.76 (2.41)	2.49	2.95
	15	91.89 (2.11)	2.18	5.31	15	87.79 (2.85)	3.85	3.50	15	92.75 (3.11)	3.25	3.76
	30 ^γ^	93.09 (1.75)	1.79	3.17	30 ^γ^	89.37 (2.99)	3.13	3.26	30 ^γ^	94.47 (1.82)	1.75	2.14
	60	93.13 (2.18)	3.98	4.42	60	91.91 (3.40)	3.42	3.57	60	91.51 (1.57)	2.44	3.70
ABZ	26.6	89.93 (2.49)	2.55	2.89	28.4	89.81 (2.00)	3.80	3.99	25.0	89.41 (2.74)	2.64	2.86
	50	91.50 (2.06)	1.95	2.80	50	88.95 (1.55)	2.49	3.30	50	95.37 (2.95)	2.43	3.46
	100 ^γ^	93.74 (2.77)	1.74	2.32	100 ^γ^	91.58 (2.83)	2.86	4.32	100 ^γ^	88.91 (2.70)	2.47	2.89
	200	93.11 (3.01)	3.16	3.57	200	90.76 (3.09)	3.01	5.07	200	94.45 (3.38)	3.13	3.68
ABZSO_2_	21.3	91.17 (3.82)	2.91	4.85	22.1	87.76 (3.01)	3.06	2.90	20.0	94.48 (3.08)	3.12	3.80
	50	90.24 (1.28)	3.34	3.86	50	89.42 (1.40)	2.29	1.74	50	91.71 (2.51)	3.40	3.62
	100 ^γ^	93.83 (2.99)	3.02	3.57	100 ^γ^	89.71 (3.21)	3.31	3.89	100 ^γ^	96.39 (2.99)	4.40	4.97
	200	91.85 (2.85)	2.86	3.81	200	90.66 (2.17)	1.91	3.05	200	94.79 (3.72)	3.69	3.57
ABZSO	7.9	91.52 (2.37)	3.32	3.26	8.1	89.96 (2.72)	2.79	3.10	7.8	90.67 (3.91)	3.86	4.60
	50	94.62 (1.64)	2.62	2.66	50	95.97 (1.69)	3.77	3.83	50	96.23 (1.86)	1.89	2.95
	100 ^γ^	93.58 (2.68)	2.83	3.16	100 ^γ^	94.63 (3.77)	3.98	4.22	100 ^γ^	95.03 (3.72)	3.61	4.16
	200	93.67 (3.35)	3.17	5.06	200	94.84 (1.79)	1.68	2.99	200	94.19 (2.49)	2.46	2.77
ABZSO_2_NH_2_	12.4	90.90 (2.84)	2.79	3.65	10.6	90.94 (1.89)	2.94	4.29	11.5	93.50 (3.10)	3.09	3.29
	50	92.57 (1.68)	1.76	2.49	50	94.00 (2.65)	3.73	2.90	50	95.42 (3.06)	3.87	4.90
	100 ^γ^	90.13 (3.19)	3.28	3.99	100 ^γ^	90.99 (3.43)	3.36	4.14	100 ^γ^	93.42 (3.81)	3.61	3.98
	200	94.58 (2.66)	2.52	2.47	200	95.57 (3.96)	4.15	4.33	200	97.21 (1.92)	2.94	2.06

Note: α, *n* = 6, values in parentheses are SDs. β, *n* = 6. γ, MRL.

**Table 5 foods-11-03894-t005:** Comparison with the reported LC methods for the detection of AVMs and BMZs in foods of animal origin.

Analytes	Sample	Sample Preparation	LC Conditions	Detection Method	Sensitivity (µg/kg)	Recovery (%)	Analysis Time (min)
Four AVMs including ABM and IVM	Ovine muscle	QuEChERS extraction, d-SPE clean-up with C18 cartridge	MeCN/tetrahydrofuran/MeOH (96:3:1, *v*/*v*/*v*)	HPLC–FLD	LOD: ABM, 5.80, IVM, 4.00 LOQ: ABM, 8.70, IVM, 5.90	100.40–121.50	>14.0 [17]
Four AVMs including ABM and IVM	Bovine liver	Extraction with isooctane, SPE clean-up with alumina-N cartridge	MeOH/MeCN/1% triethylamine and 1% phosphoric acid (61:30:9, *v*/*v*/*v*)	HPLC–FLD	LODs: - LOQs: 2.00	84.00–96.00	>19.7 [14]
Four AVMs including ABM and IVM	Bovine liver	Extraction with MeOH, SPE clean-up with immunoaffinity cartridge	MeOH/water (98:2, *v*/*v*)	HPLC–FLD	LODs: - LOQs: 2.00	79.30−115. 90	30.0 [18]
Four AVMs including ABM and IVM	Milk	Extraction with MeCN	MeCN/tetrahydrofuran/water (90:6:4, *v*/*v*/*v*)	HPLC–FLD	LODs: 0.30 µg/L LOQs: -	83.50–93.70	20.0 [19]
Three AVMs including ABM and IVM	Bovine liver	Extraction with MeCN, SPE clean-up with aluminum B cartridge	-	HPLC–FLD	LODs: - LOQs: 1.00	72.00–81.00	20.0 [20]
Two BMZs including ABZ	Milk	Dispersive liquid phase microextraction-solidified floating organic drop	MeOH/water (80:20, *v*/*v*)	HPLC–FLD	LOD: ABZ, 0.02 µg/L LOQ: -	96.00–104.30	12.0 [32]
ABZ and three metabolites	Fish muscle with adhering skin	Extraction with ethyl acetate	MeOH/MeCN/0.025 mol/L ammonium acetate (12:8:80, *v*/*v*/*v*)	HPLC–FLD	LODs: 0.20–3.00 LOQs: 0.70–11.00	65.00–108.00	35.0 [15]
ABZ and three metabolites	Fish muscle	Extraction with ethyl acetate	MeCN/MeOH/0.05 mol/L ammonium acetate (30:15:55, *v*/*v*/*v*)	HPLC–FLD	-	67.00–94.00	>17.0 [16]
ABZ and three metabolites	Pig and poultry muscle	Extraction with ethyl acetate, SPE clean-up with Oasis PRiME hydrophilic-lipophilic balance cartridge	MeCN/aqueous solution (containing 0.2% formic acid and 0.05% triethylamine) (31:69, *v*/*v*)	UPLC–FLD	LODs: 0.20–3.80 LOQs: 1.00–10.90	80.37–98.39	6.0 [4]
Ten BMZs including ABZ and three metabolites	Total egg	Extraction with MeCN	MeCN, 0.025 mol/L ammonium acetate (pH 5)	HPLC–UVD	LODs: 5.00–134.0 LOQs: 100.0–250.0	68.90–98.30	25.0 [21]
Three BMZs including ABZ	Milk	Solid phase microextraction	MeCN, 0.1% formic acid	HPLC–PDAD	LOD: ABZ, 0.11 µg/L LOQ: ABZ, 0.70 µg/L	72.30–121.00	21.0 [23]
Four BMZs including ABZ	Liver (chicken, pig, and bovine) and kidney (chicken and pig)	Vortex-assisted surfactant-enhanced emulsification microextraction with solidification of floating organic droplet	MeOH, 1% acetic acid	HPLC–PDAD	LODs: 0.03–0.05 LOQs: 0.10–0.20	87.00–105.00	15.0 [26]
Five BMZs including ABZ	Milk	Ultrasound-assisted cloud-point extraction	MeOH, 1% acetic acid	HPLC–PDAD	LODs: 0.005−0.10 µg/L LOQs: -	75.30–111.40	18.0 [27]
Five BMZs including ABZ	Milk	Ultrasound-assisted surfactant-enhanced emulsification microextraction	MeOH, 1% acetic acid	HPLC–PDAD	LODs: 1.80–3.60 µg/L LOQs: 5.30–11.00 µg/L	72.50–113.50	18.0 [28]
Four BMZs including ABZ	Milk	Surfactant-solvent-based quaternary component emulsification microextraction	MeCN, 0.1% formic acid	HPLC–PDAD	LODs: 2.60–9.90 µg/L LOQs: -	80.10–114.10	9.0 [24]
Four BMZs including ABZ	Egg	QuEChERS, ultrasound- assisted emulsification microextraction	MeCN, 0.1% formic acid	HPLC–PDAD	LODs: 7.20–14.40 LOQs: -	74.30–112.90	8.0 [25]
ABM, IVM, ABZ and three metabolites	Egg	Extraction with MeCN/water (90:10, *v*/*v*)	MeOH, 0.1% TFA	HPLC–PDAD	LODs: 2.10–10.50 LOQs: 7.80–28.40	85.70–97.21	11.0 [This study]

Note: -, not provided.

## Data Availability

All available data are contained within the article.

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
