# Peer review of "Separation and Detection of Abamectin, Ivermectin, Albendazole and Three Metabolites in Eggs Using Reversed-Phase HPLC Coupled with a Photo Diode Array Detector"

_foods, 2022, doi:10.3390/foods11233894_

Round 1
Reviewer 1 Report
The authors should make some changes to the documents:
1) section 2.3 is the 2 weeks of feeding with nonmedicated feed a standard?
2) consider rewriting the abstract. "and adaptable by being adopted..."
3) have you considered using an internal standard based on the complex matrix
4) please include the structures of the analytes of interest
5) there is no need for having the chromatogram a) mixed standards 3 times in Figures 1-3.
6) since different wave lengths have been used in the methods, please include the wavelength for the Figure 1-3. please add in the supplementary the chromatograms at different wavelenghts.
7) what is th disturbance in the baseline at 7-8 minutes? you state at line 287 that the baseline is stable.
since this is a chromatogram at some higher wavelenghts, the solvent should not show this kind of effect for the gradient.
8) please add how the LOD and LOQ were determined/calculated
9) is the precistion (line 324) determined for retention time or for peak area?
10) are method times available for methods in Table 5 since you point out your method is time saving.
11) please comment the obtained linear range in terms of expected concentrations in real samples.
12) is the self-reference No. 9 really necesary and the best example for the sentence
Author Response
Thank you for your comments and suggestions on this manuscript. We comprehensively revised and improved the manuscript based on your professional, accurate and detailed advice. We hope that our responses satisfy you and that the revised version meets your requirements.

Reviewer 2 Report
The manuscript is well structured. The title clearly describes the contents of the paper.
However, some additions need to be made.
Why is the photodiode array detector important? These compounds can also be detected with other detectors.
How are these extractions different from previous extractions?
I think it would be better to make some additions on these issues.
Author Response
Thank you very much for carefully reviewing our manuscript and giving some positive comments on the study. We are truly grateful for your critical comments and thoughtful suggestions. Based on your suggestions, we have carefully studied some high-quality literature and revised our manuscript.

Reviewer 3 Report
The introduction has to be rewritten. Provide information about the frequency of application of the tested compounds and their biotransformation The methodology should describe the validation of the applied methods of determination in more detail.
The results should be statistically analyzed
Author Response
We thank you for your valuable comments and solutions, which provided direct and effective suggestions for improving this manuscript. We are aware of the shortcomings of the manuscript, and we hope that the revised manuscript has satisfactorily addressed the reviewer’s concerns.

Round 2
Reviewer 1 Report
Dear authors,
you have made substantiall changes to your manuscript based on the comments of the reviewers.
However I have to again raise two points:
point 9) What you describe as precision „The precision was determined on the basis of the recovery, which was calculated by substituting the peak area into the corresponding linear equation, as explained in section 2.5.” this in fact is the precision of the method. Please have a look at FDA, EMA or ICH guidelines for analytical method validation.
point 11) but when you developed your method you aimed for some working concentration range. You chose this concentration range based on some expected/possible concentrations in the real samples.
You state – the real samples did not contain the targets at detectable levels. How do you know that this is just not an issue of your developed method and that its limits of identification/quantification are just not low enough.
Author Response
We hope our responses are clear and unambiguous, and we hope this discussion will not cause doubts or dissatisfaction. We greatly appreciate your professionalism and meticulousness.
